

# Where is all the research software? An analysis of software in UK academic repositories

Domhnall Carlin[1], Austen Rainer[2] and David Wilson[1]

[1] Institute of Electronics, Communications and Information Technology, The Queen's University Belfast, Belfast, Northern Ireland, United Kingdom
[2] School of Electronics, Electrical Engineering and Computer Science, The Queen's University Belfast, Belfast, Northern Ireland, United Kingdom

## ABSTRACT

This research examines the prevalence of research software as independent records of output within UK academic institutional repositories (IRs). There has been a steep decline in numbers of research software submissions to the UK's Research Excellence Framework from 2008 to 2021, but there has been no investigation into whether and how the official academic IRs have affected the low return rates. In what we believe to be the first such census of its kind, we queried the 182 online repositories of 157 UK universities. Our findings show that the prevalence of software within UK Academic IRs is incredibly low. Fewer than 28% contain software as recognised academic output. Of greater concern, we found that over 63% of repositories do not currently record software as a type of research output and that several Universities appeared to have removed software as a defined type from default settings of their repository. We also explored potential correlations, such as being a member of the Russell group, but found no correlation between these metadata and prevalence of records of software. Finally, we discuss the implications of these findings with regards to the lack of recognition of software as a discrete research output in institutions, despite the opposite being mandated by funders, and we make recommendations for changes in policies and operating procedures.

## INTRODUCTION

Research software engineers (RSEs) apply professional practises to the development of software, where the end goal of the activity is not actually the software itself, but rather a tangible research output (*e.g.,* a journal article, a conference paper etc.). If made well, the software then also becomes a tangible research output, eligible for a Digital Object Identifier (DOI), publication and reuse to underpin other research. However, despite growing recognition of the importance of research software, the developers—or RSEs—are rarely credited either through citation (*Park & Wolfram, 2019*) or recognition of the output (*Struck, 2018*). There is a sustained international effort to gain recognition for software artefacts that are essential in producing the results that underlie published articles.

Corresponding author
Domhnall Carlin, d.carlin@qub.ac.uk

To enable this, research software has to be published *formally* (*Druskat et al., 2022*). Indeed, Jisc (previously JISC- Joint Information Systems Committee), which provides IT and networking infrastructure to the UK's academic institutions (*Jisc, 2021*), makes the following recommendations:

> 'Treat computer code like any other output of your research... Share your computer code like you would any other research output...Computer code should have a URL or a DOI (digital object identifier)... Always include these when citing the code, including information on the version you used.' (Research Data Management Toolkit, 2021)

The open sharing of artefacts of research provides a variety of benefits for stakeholders in the research ecosystem. It enables the auditing of claims made in research outputs through reproduction (*i.e.*, same input data and process should equal the same output) and replication (*i.e.*, different data and same process should yield an equivalent output). Trust in the results of a paper comes from the peer review system, but it is difficult to verify claims when they rely on unpublished code (*Hasselbring et al., 2020a*). Open sharing enables sustainability, and extensibility, through the reuse of both the code itself and knowledge potential of the completed research, rather than allowing the code to "languish", *e.g.*, on a USB stick in a desk drawer. When shared openly, research software can also help minimise development time, reduce duplication and minimise repeated effort (*Thelwall & Kousha, 2016*).

Despite the benefits of open sharing of software, there has been a serious decline in returns of software as exemplar research outputs in their own right. REF, the UK's Research Excellence Framework, aims to periodically assess higher education research quality and impact. In REF2014, 97% of all research outputs were 'publications' or 'books', up from 94% in the 2008 equivalent and this has risen to 98% in the current REF2021 (*Cleaver, Derrick & Hettrick, 2022*). Digital artefacts, including software, have fallen from 5,252 submissions in 2008, to 761 in 2014 (*Research Excellence Framework, 2014*) to 469 in 2021 In 2008, 113 (2%) of the 5,252 digital artefact submissions were software (*Hettrick, 2022*), yet this has declined in number to 38 in 2014 (*Research Excellence Framework, 2014*) and to 11 in 2021 (*Research Excellence Framework, 2021*). As a contributing percentage to all digital artefact submissions, software has fluctuated from 2.15% to 4.99% and back to 2.35%. However, as a percentage of total returns, software has declined nearly tenfold from 2008's level (0.052%) to 2021 (0.006%).

Following this trend of REF results, the questions being asked in the RSE community are: what is causing this remarkable drop in software returns, especially considering the sizeable expansion of the RSE movement in the UK?, and what can be done to improve returns of software in the REF? To examine the various strands from a user perspective, we have separated out the use case for researchers to publish their software into three statements: As a researcher, I should (i) know I can publish my research software, (ii) within my institution's guidelines, (iii) with my institution's research repository. The present work examines the third strand, seeking to examine the extent of software as research outputs in UK academia.

In this paper, we query 182 institutional repositories to take the first such census of Research Software in UK Academic Institutional Repositories. Previous work in the field has largely focussed on the role of research software in academic output through the role of the Research Software Engineer, the correct assignment of output credit and workflows for tying code to datasets and papers to underpin good Open Science. While there has been some discussion of REF returns for software in the UK, there has been no attempt to investigate the role of the institutional repository in the low return rates. Given the imperative nature of the institutional repository in terms of meeting Open Access requirements and thus REF-returnability, it must be an essential piece in the workflows of Open Science. However, our findings show (i) that the prevalence of software, represented by independent records of output, within UK Academic Institutional Repositories is incredibly low, with fewer than 1/4 containing software as academic output; (ii) that over 2/3 of repositories *cannot* contain software as a defined type, as they *do not* recognise it as a defined type of output; (iii) that a number of universities actively remove the ability to deposit software as a defined type from their repository, based on deviations from the default settings of their platform; (iv) that no correlation could be found between metadata describing the infrastructure, the institution and the presence of RSE groups. Lastly, we discuss the implications of these findings with regards to the lack of recognition of software as discrete research output in institutions, despite the opposite being mandated by funders, and we make recommendations for changes in policies an operating procedures.

The remainder of this article is structured as follows. Section 'Background and Wider Context' provides information on background and wider context of software as research output and repository infrastructure. Section 'Related Work' presents related research and defines the research questions for the present work. Section 'Method' defines the methods used to obtain the census data, including those that failed. The results of the census and further analysis are presented in Section 'Results', which are further discussed in Section 'Discussion'. Finally, Sections 'Conclusions' and 'Recommendations and Future Work' present the conclusions from the research and make several recommendations for policy change, practical workflows and future research directions.

## BACKGROUND AND WIDER CONTEXT

This section presents fundamental information on research software as valid scientific output, the IRs that store such outputs and the metadata used to describe them. These factors prove important in determining whether a repository can actually index a piece of research software, as discussed later in the present work. Here we also present background information on other variables we anticipated could correlate to the presence of software deposited in a given IR.

### Software: a definition

What exactly do we mean by *software* in the context of research? The COAR Controlled Vocabularies for Repositories defines a *resource type* named *software* as 'A computer program in source code (text) or compiled form' (*COAR, 2022a*), which is sourced directly from the DCMI term (*DCMI, 2020*) and similar to the US Library of Congress's 'computer

program' (*Library of Congress, 2021*). More specifically, both *research software* (*COAR, 2022b*) and *source code* (*COAR, 2022c*) are defined as specific entities in and of themselves within this vocabulary. *Schema.org (2023)* a community body established by Google, Microsoft, Yahoo and Yandex, seeks to define schemas for structuring data on the internet. This too includes an entity of *software application*, defined as a subclass of a *creative work*, which is in turn a subclass of a '*thing*'.

## Software as research output

REF, which aims to assess UK higher education research quality and impact, describes research outputs as the 'published or publicly available products of research, which can take many forms. These include ... software' (*Research Excellence Framework, 2022*). *Sompel et al. (2004)* enumerate the four functions of scientific communication as proposed by *Roosendaal & Geurts (1998)* and append a fifth:

1. Registration: the claim of precedence of a research finding is made.
2. Certification: the validity of the registered claim is established *e.g.*, peer review.
3. Awareness: the registered claim is disseminated to others.
4. Archiving: the record of the registered claim is preserved.
5. Rewarding: the participants in the communication system somehow benefit from derived metrics.

*Sompel et al. (2004)* go on to say:

> 'The system should consider datasets, simulations, software, and dynamic knowledge representations as units of communication in their own right.'

It is hard to find where research software does not fit with these five function statements as part of the overall purpose and ethos of scientific communication. However, there have been claims of reproducibility crises across science for several years, with particular emphasis on the components of the research process, including research software. For example, *Baker (2016)* surveyed 1,500 scientists regarding reproducibility within their research fields. When asked what factors contribute to irreproducible research, over 81% said that methods or code being unavailable 'always' (13.8%), 'very often' (31.6%), or 'sometimes' (36.3%) contributes to the problem.

Specific journals exist to peer-review research software and publish it (*e.g.*, Journal of Open Source Software, Journal of Open Research Software, Software Impacts etc.), along with a growing number of 'artefacts tracks' at large conferences. Prestigious conferences from organisations such as IEEE, USENIX and ACM have begun adding 'badges' to deposited software within these tracks, awarding levels of reusability based on the code quality. Publishing code under this peer review process forces the author to write better code to a higher standard. Ultimately, better software means better research: greater research integrity, reproducibility and rigour.

Software is specifically mentioned by the San Francisco Declaration on Research Assessment, known as DORA (*Declaration on Research Assessment, 2012*) as a legitimate research output, with a recommendation to research institutions and publishers that "For the purposes of research assessment, consider the value and impact of all research

outputs (including datasets and software) in addition to research publications.''(DORA, Recommendations 3 & 5).

UK Research and Innovation (UKRI) is the UK's national funding agency, tasked with investing over £7.9 billion annually in science and research across the UK. The body also sets out terms and requirements for research outputs supported by its funding, including the management of Open Access (OA) publications and materials that support research. Their OA Policy (*UKRI Open Research Team, 2022*) states that, in order to meet the *Concordat on Research Data*:

> "...it is a requirement for in-scope research articles to contain a data access statement. This informs readers where the underlying research materials associated with a paper are available...Underlying research materials are research data, as defined in the Concordat on Open Research Data, and can include code, software... "(*UKRI Open Research Team, 2022*, pp. 10).

The Concordat (*UKRI, 2016*) lays out principles to ensure that the research data yielded from UK research is made openly available for use by others wherever possible. It notably includes reference to not only the data that underpins a publication, but the software required to generate the results of the output:

> "...it is vital that the data supporting and underlying published research findings should, as far as possible, be made open by the time the findings are published and be preserved for an appropriate period. This could be achieved by depositing and providing access to relevant data and associated software (where possible) via a repository owned or operated by a discipline-specific research community and its funding bodies, a publisher, a research institution, a subject association, a learned society, national deposit libraries or a commercial organisation; or via other mechanisms that provide appropriate and sustainable services." (UKRI Concordat on Research Data, Principle 8, pp 16)

In essence, this concordat requires that the research software must be correctly stored and made accessible for UKRI-funded research, in order to allow for reproducibility, replication and sustainable research. In the 20th anniversary update (*BOAI, 2022*) to the 2002 Budapest Open Access Initiative (BOAI) (*BOAI, 2002*), the authors added an explicit statement on how open access research software should be stored, with particular reference to the risk posed by employing commercial platforms, such as GitHub, to do so:

> "Host OA research on open infrastructure. Host and publish OA texts, data, metadata, code, and other digital research outputs on open, community-controlled infrastructure. Use infrastructure that minimizes the risk of future access restrictions or control by commercial organizations." (*BOAI, 2022*)

Efforts have been made to promote the application of the FAIR (Findable, Accessible, Interoperable, Reusable) principles for scientific data management and stewardship (*Wilkinson et al., 2016*) to research software (*e.g.*, *Chue Hong & Barker, 2021*, *Lamprecht et al., 2020*, *Clément-Fontaine et al. 2019*). The FAIR for Research Software (FAIR4RS) working group published their FAIR Principles for Research Software v1.0 (*Chue Hong et*

*al., 2021*), a community-sourced conversion and reapplication of the FAIR principles to the specific features and requirements of research software.

One factor pertinent to this issue is the common conflation of research software with research data or methods. Some authors claim that software is not just inextricably linked to data, but that it is a form of data in and of itself (*Marcus & Menzies, 2010*). Others specify that, while technically a form of data as it can be processed by a computer, software is different as it is executable and should be considered as a creative tool that operates on data (*Katz et al., 2022*). While it is clear *why* software, and other essential components, should be retained as artefacts of the research process, the *how* of this principle remains both rather unclear and highly variable.

While the RSE movement has certainly grown rapidly within UK universities, it is not clear as to any effects this has had on the presence of software within UK academic repositories. Similarly, when investigating the role of the institutional repository in the low return REF return rates for software, differences may be found in line with the research-intensity of the institution. For example, The Russell Group is composed of 24 research-intensive universities in the UK, producing more than two-thirds of the world-leading research output of UK universities (*Russell Group, 2022*). Our investigation seeks to examine such variables as potentially correlated with the level of software within the IRs.

## Software as a REF-returnable item

As previously mentioned, the REF exercise assesses research outputs from UK Universities for quality and impact. Outputs that are eligible for submission to the institution's REF process are termed REF-returnable. Strict open access guidelines for certain outputs (*e.g.*, journal articles) are in place to maintain REF-returnability, which partly drove the rapid adoption of IRs by UK universities. However, REF exempts non-textual outputs, including software, from this open access policy (*Research Excellence Framework, 2019*). The REF submission guidance does state in its definition of software that it '...has been made publicly available'. A written description of the software must be provided, along with how the software (including source code if applicable) can be accessed (*Research Excellence Framework, 2019*, pp. 110). Mandating that the software is 'publicly available' is rather ambiguous and could be met by a wide variety of solutions, not necessarily indexed *e.g.*, Google Drive is publicly available with the correct link, but could not be described as publicly *discoverable*, which a repository can facilitate. This suggests that there is no requirement for software to be deposited in the IR to be REF-returnable. However, part of the *institutional* management process for REF might only take consideration of items deposited in the IR. For example, Queen's University Belfast guidance states that a module within the IR, using the Pure platform, is responsible for administering the selection process for the institution's REF return, suggesting that only items in the IR are 'visible' to the process (*Queen's University Belfast, 2019*, p. 4.4.2).

## Repositories

Since the milestone manifesto of *BOAI (2002)*, the Open Access movement has transformed the age-old faculty tradition of self-archiving research outputs into the *de rigueur* workflow

for academia. With funders beginning to mandate open access requirements for research, the institutional repository quickly became the essential self-archiving platform. *Crow (2002)* define the institutional repository as:

> "...digital collections capturing and preserving the intellectual output of a single or multi-university community..."(*Crow, 2002*, pp. 1)

*Institutional* repositories, typically called Research Information Systems (RIS), contain permanent records of research output from employees of the university. While these exist largely to fulfil open access requirements for funders, they also provide metadata on their contents for services that aggregate harvested content, the primary one of which is described in the following section.

There is an important distinction to be made between types of repositories that are used by academic software developers as mentioned in the literature. *Code* repositories, such as GitHub, GitLab etc., offer collaborative development tools for open-source software *e.g.*, issue-tracking, versioning, release management. While mainly focussed on active development of software, they can accommodate archiving of code that is not under active development, but is still usable. *Archival* repositories provide accurate long term immutable references to any form of digital object, tracked by a permanent DOI. While this format enables long-term reference storage with FAIR attributes, it is not suitable for active and collaborative development of software and is targeted more towards traditional paper formats. However, some archival repositories (*e.g.*, Zenodo) are software-friendly, with interactive integrations to GitHub etc. Further to this, *Garijo et al. (2022)* make a distinction between research software *registries* and *repositories*, with the former indexing code without storing it and the latter doing both.

As described by the CodeMeta Project (*Jones et al., 2017*), the infrastructure underpinning the description, indexing, archiving and discoverability of research software is far behind that of traditional research outputs, though not for technological reasons. The issue, the authors state, is in the interoperability of disparate platforms, scientific disciplines and descriptors for research artefacts, *i.e.*, individual repositories cannot communicate in a common language to describe their data to each other. The authors cite an example of workflow integration that allows persistence for both code and state from GitHub on the former within the archival format of FigShare. However, in order to attain a DOI for the output, the pair must go to a third body (DataCite) to have one minted. The three offer differing metadata, meaning the detailed descriptors can be lost to more generic, less granular equivalents in translation. This metadata is crucial in making software FAIR, but the variance among standards seems to be holding the FAIR movement back in this regard.

## Metadata & protocols

When examining *how* types of research output are stored, it is important to examine *what* they are labelled as. This work is undertaken by the metadata format used to describe the object through a codified list of identifiers and across a variety of descriptors. Different metadata formats offer differing vocabularies, as identified above, which will ultimately determine the classification of research output into discrete categories. For

example, if a piece of research software is described under a metadata format without 'software' as a specific type of output, it cannot be described as 'software' using that format and will typically be referred to as 'other'. The Dublin Core™ Metadata Element Set (commonly referred to as the Dublin Core or DC) is the metadata format, decided upon with international consensus, that has been formally standardised as ANSI/NISO Z39.85 and ISO 15836. It contains 15 core properties to describe resources. The DC label 'Type' is defined as the 'nature or genre of the resource' (*DCMI, 2020a*) and it is recommended to use a fixed or controlled vocabulary to define types, such as the DCMI type vocabulary, which includes a fixed type named Software (*DCMI, 2020b*). This is an important point to note, as many repositories using metadata formats that inherit from DC actually do not include software as a defined type of output, as discussed later in the paper.

The Research Information Systems (.ris) file format is a standard text-based tagged template for bibliographic information, including a fixed vocabulary of 35 types of publication, one of which is the *COMP (Computer Program)* type of reference (*RIS File Format, 2012*). While this is a very common format, supported by popular digital libraries and reference managers, there is no obligation on the importing agent to enforce tags of types. For example, it does not appear to be accepted by resource list management platform Talis Aspire, which omits it from a list recognised types (*Hodson, 2019*) and states non-conforming types will be rejected from import.

RIOXX: The Research Outputs Metadata Schema was originally developed to meet the then-newly created reporting requirements of Research Councils UK (now UKRI). The schema was designed to allow institutional repositories to share metadata and enable consistency in the tracking of items of open access research outputs across disparate systems. Version 1 appears to have allowed free text with the 'type' description (*Rioxx: The Research Outputs Metadata Schema, 2023a*). The current v2 (*Rioxx: The Research Outputs Metadata Schema, 2023b*) mandates one of 14 types from a controlled vocabulary, none of which include software or similar, meaning repositories that use the RIOXX format cannot explicitly label software by type. However, the draft third version of the metadata format has a proposal to include the vocabulary of COAR within its own type category (*Walk & Brown, 2023*).

Jisc maintains the Institutional Repository Usage Statistics (IRUS) service, which allows users to gather and share comparative statistics on research item usage. This service did not include software as a type in the original definition of IRUS item types from 2012. This omission was maintained through two further updates in 2015 and 2020, until the most recent definition in August 2022. Jisc uses the DC Type fields, which include software, but maintains its own vocabulary. Why, then, was software not counted within this service originally? Jisc explains in great detail the evolution of their policies on item types in (*IRUS-UK, 2022*). The original 25 item types were distilled from over 700 types identified by the Intute Repository Search (IRS) project (*Jisc, 2016*). This project mapped all item types in use in UK repositories, which included software (*Reed, 2014*), and the most populous were selected. Jisc's report also illustrates the variance and inconsistency in item types, noting that DCMITYPE (DSpace), Sympletic, Pure, Jisc Infokit List 1, CERIF and

REF2014 all included the type *Software*, with *Code* recommended by Schema, though these were mapped to *Other* in the iterations up until August 2022. The report states:

> "In addition, we are starting a pending list of item types that we are considering for inclusion in the future if their usage becomes significant. We have been asked to consider the inclusion of 'Code' as an item type. None of our current participants uses that but some use the term 'Software' so we have added this to our pending list." (*Reed, 2014*, p. 13)

It appears that the attempts to standardise metadata for research outputs have not always been adequate for consideration of software as a legitimate output, either in how they were designed or in how they have been implemented.

**The open archives initiative protocol for metadata harvesting (OAI-PMH)**
The OAI-PMH protocol was designed to provide a framework to enable metadata harvesting between online repositories, e-print servers, digital libraries etc., in a manner that is application-independent and applicable across many archive types. The first version of the specification of the OAI-PMH interoperability architecture was released to the public in January 2001 (*OAI, 2001*), with 2002′s v2 being the current implementation. It uses XML over HTTP, with an XML schema associated with the OAI_DC metadata prefix (*Johnston, 2002*). Each implementation of OAI-PMH *must* support the Dublin Core metadata format as a minimum, but may also support other formats too.

OAI-PMH also allows selective harvesting based on 'sets', *i.e.*, preconfigured collections of records that are grouped together along some common attribute. The sets for each OAI-PMH server can be determined with a simple API call, which must be implemented for all OAI-PMH instances. However, IRs are not obligated to actually use sets and can return empty lists. This is discussed further in the methodology section below.

## RELATED WORK

As a result of the evolution of researchers-who-write-code into RSEs, the standard practice of employing publications to assess an individual's academic performance has continued from academia. This often means that RSEs must write software and publish papers to prove their value under this credit system (*Sochat, 2022*), with citeability being a prerequisite to recognition and progression (*Struck, 2018*).

*Hasselbring et al. (2020b)* analysed the relationship between research software and publications by examining both publication and software metadata in repositories. Over 5,000 GitHub repositories were identified as research software because they either referenced a parent paper (*e.g.*, through DOI) or were themselves referenced in a research paper from arXiv or the Association for Computing Machinery (ACM) Digital Library. The authors note five observed relationships between research publications and research software:

1. 'software as an output of research, collaboratively constructed and maintained through an active open source community;'

2. 'software as an output of research, privately developed but published openly and abandoned after dissemination;'
3. 'software itself as an object of study or analysis;'
4. 'software that leads to a fork (in GitHub) that is independently developed as a research output and published openly (if successful, it may be fed back into the original project via GitHub pull requests);'
5. 'software used as a tool or framework to perform the research.' (*Hasselbring et al., 2020a*; *Hasselbring et al., 2020b*, pp. 87)

The authors go on to suggest a combination of Zenodo and GitHub to meet archival and maintenance requirements, though do not mention institutional repositories.

*Druskat & Katz (2018)* sought to map the actors and actions in the research software space in a more granular version of *Katz (2018)*. The authors differentiate between code repositories and archival repositories, but fail to mention institutional repositories. Research software and its metadata are spread over many repositories, including major and highly populated and smaller repositories with varying focuses (*Howison, Conklin & Crowston, 2006*). *Park & Wolfram (2019)* analysed over 67,000 research software records across public repositories indexed by Clarivate Analytics Data Citation Index. They found that research software was rarely cited, with 7,099 total citations equating to an average of 0.1 citations per record. The authors note that the informal citation of software by writers (*e.g.*, footnotes rather than full reference) and lack of indexing of these informal approaches may be an important factor.

*Hasselbring et al. (2020a)* present practical discussions on the suitability of various repositories, and combinations thereof, for allowing research software to meet FAIR principles. While software-friendly archival repositories like Zenodo can interoperate with development-focussed repositories like GitHub to meet these principles, there is a missing link with records in the ultimately imperative institutional repository.

*Garijo et al. (2022)* reported on the work of a task force (part of the FORCE11 Working Group) to examine best practices for research software repositories (*i.e.*, code storage with metadata) and registries (*i.e.*, records only containing metadata and links to repositories). The work of the task force included an analysis of 14 software repositories, examining if they were: active; accepted software; could mint DOIs; could be used to cite software; and if they were discipline-specific. This research found that most of the 14 resources do accept a software deposit, support DOIs, are capable of being used to cite their contained software and are curated actively. While the repositories and registries are not explicitly listed, the work does not appear to mention the idea of academic institutional repositories being used for permanent records of metadata.

*Struck (2018)* gave an overview of research software discovery processes and tools. The authors highlight the popularity of GitHub for hosting research software, but point out that maintainers of institutional repositories are sometimes reticent in permitting external authentication methods, *i.e.*, not wishing to move official services to externally authenticated hosts. The authors suggest that this could hamper collaboration on research software and its sharing or reuse, potentially pushing users to commercial solutions.

*Wattanakriengkrai et al. (2022)* analysed the traceability between networks of academic papers and software repositories on GitHub. The authors examined 20,000 GitHub repositories that reference an academic paper (typically through the README.md). Over 50% of these repositories were implementations of methodology or algorithms in someone else's paper, with 40% being references to the repository owners' own academic paper. To examine the relationship from papers to repositories, a sample of 2,032 academic papers from seven of the top prestigious publication venues in software engineering were analysed and the authors found that these do not typically reference a repository, but it is normally GitHub if at all. In the analysis, it does not appear that any article references or links back to an institutional repository, with only five public repositories making up the entire distribution.

From the literature, it appears that research into the depositing of research software does not generally consider institutional repositories in their analyses. In terms of research methodology and subsequent recommendations, the open access repositories maintained by universities seem to be considered separate from other types.

To the best of our knowledge, with the exception of the IRS project (*Jisc, 2016*) no existing research has performed a census of research software contained in academic institutional repositories. However, the IRS project's focus was to improve search capabilities for the fledgling open access repository movement from 2007-9, with the recording of publication types a byproduct of this work. All of the literature reviewed points to the necessity of research software for open, reproducible and sustainable research. The lack of academic credit for research software, which drove the RSE movement, is often mentioned. Many make recommendations for dual archiving of research software to achieve persistence and maintenance. However, none of this body of research joins these tenets, specifically with regard to the recording of research software as entries in institutional repositories, which is essential in the current assessment of research excellence.

This article seeks to answer the following research questions:

RQ1: to what extent is software published in academic repositories?

RQ2: are there any explanatory variables associated with whether software is included in a repository?

RQ3: are there barriers preventing repositories from storing records of software as distinct research outputs?

## METHOD

Here, we present the methodology used to generate our experimental dataset and conduct the analyses. For completeness and to guide future work, we also include the aspects of our methodology that failed to yield data that was robust and comprehensive enough. To briefly summarise the approach employed, our initial effort was to investigate existing search services within repository aggregation websites for the data on the presence of software in IRs, which could not be attained. Next, we gathered a list of IR URLs from various aggregation sites, in order to target specific searches of the IRs for the presence of software. Using the OAI-PMH protocol, we ran queries against all available IR URLs

[1] Query: https://share.osf.io/discover?q=%20tags%3A(%22Software%22)

[2] Query: https://explore.openaire.eu/search/find/research-outcomes?type=%22software%22

[3] Query: https://www.base-search.net/Search/Results?lookfor=country%3Auk+doctype%3A6&l=en&oaboost=1&ling=0&newsearch=1&refid=dcadven&name=

[4] Query: https://v2.sherpa.ac.uk/cgi/search/repository/advanced?screen=Search&repository_name_merge=ALL&repository_name=&repository_org_name_merge=ALL&repository_org_name=&type=institutional&content_types=software&content_types_merge=ANY&content_subjects_merge=ANY&org_country_browse_merge=ALL&org_country_browse=United+Kingdom&satisfyall=ALL&order=preferred_name&_action_search=Search

to detect software, using the selective harvesting approach to search for sets indicating software as a type of output. This approach failed to provide accurate results and so a full manual check was made by visiting each IR URL through a web browser. Although laborious, this provided a ground-truth census dataset that enabled analyses to uncover correlations between descriptive variables and the presence of software. It also enabled the testing of further research software designed to perform these searches in an automated fashion, by providing expected data on the number of software records present.

## Preliminary search

Several services exist to ingest, correlate and index metadata on research outputs from a wide variety of sources. The key starting point in our census was to examine search-enabled registries and repositories as central data stores for the existence of research software. While this approach is limited by the fact that these searches index more than institutional repositories, it can provide an indication of the wider approach to software depositing.

The Open Science Framework (OSF) shows 6,146 works[1] globally with the tag 'software', out of a total collection of 4.7 million. Open Access Infrastructure for Research in Europe (OpenAIRE.eu), which enables compliance with the European Commission open access policy, has specific categories for research software along with datasets and publications. It lists 688 pieces of research software from the UK out of 304,376 listed globally. The total record of all 'research products' is currently over 162.4 million, with 5.6 million attributed to the UK[2]. The Bielefeld Academic Search Engine (BASE) lists 3,165 hits from 313.8 million documents for software in the UK, though 2,204 are from BioMedCentral[3]. While the search engine has a specific selector for software, the records also include items such as digitally-held video documentaries. This highlights an issue with categorisation from primary sources when picked up by harvesting entities, which is discussed further in the 'Background and wider context: Software as research output' subsection. CORE.ac.uk is the UK aggregator for research outputs (Knoth & Zdrahal, 2012), operated as a not-for-profit service delivered by the Open University. CORE is based on a collection of 243 million searchable research 'papers' (CORE, 2022). Within CORE's search API, the 'type' parameter is limited to only 'Research', 'Thesis', 'Unknown' and 'Slides'. This again highlights the issue of missing, erroneous or incomplete metadata from the source records. CORE includes RIOXX v2 as a metadata profile, which again does not allow software as a type. Further, the difference between the software as a record in and of itself and the more traditional 'paper presenting software' is not clear cut in many cases. The OpenDOAR search service from Jisc allows detailed searches to be made of contributory repositories, including per repository type and filtering to include those that contain software. When queried for institutional repositories that list software, 23 IRs were returned[4]. The returned IRs did not include some known to contain software (e.g., our own institution was not included) and so this information was deemed incomplete.

While these searches were comprehensive in many ways, they did not provide the quantity, quality or granularity necessary for the current investigation and a full specific search of UK institutional repositories was required, as described in the following section.

[5]Query: https://explore.openaire.eu/search/content-providers?datasourcetypename=Institutional%20Repository&datasourceodlanguages=English&country=GB

## Surveying academic institutional repositories
### Data Sources

Before surveying academic repositories, it was necessary to compile an accurate list of URLs for UK academic IRs. Initially, several existing aggregating or indexing sites were consulted for a comprehensive list of IRs, with the following results. OpenAIRE.eu provides a full list of data providers to their service. When filtered to the UK (country code GB) and institutional repositories, only 50 remained. These were retained for verification, but did not provide an adequate list of UK academic institutions[5] . Contributory data sources were acquired from OAI (http://www.openarchives.org/Register/BrowseSites), which were filtered for '.ac.uk' URLs only, leaving 101 institutions with corresponding OAI URLs. However, many of these were outdated and erroneous and a more comprehensive list was required. Using the API for CORE, the API request 'searchdata-providers' was made with parameter `location.countryCode:gb`. This gave 247 responses, which was further filtered to only include .ac.uk and a small number of other domains (*e.g.*, *.worktribe.com), providing 167 total URLs from 149 unique second-level domains (*e.g.*, *.qub.ac.uk) to begin searching, along with descriptive data about each endpoint. Importantly, this indicates that one institution can have more than one repository. While extensive, this list only included contributors to the CORE aggregation system and the list was further enhanced by cross-checking with OpenDOAR's list of providers, bringing the final list to 182 repositories from 157 institutions. However, verifying this figure for completeness proved to be less than straightforward. In the UK, education is devolved to the four constituent nations, each of which takes a different approach to categorising registered tertiary education providers. The Academic Institutions we refer to encompass Further and Higher Education providers, some of which have degree-granting powers, both taught and research, and from Foundation degree (pre-undergraduate) to Doctoral degree. Including only those institutions using the title 'university' would omit many institutions, as some may legally use the word 'university' in their title while others may not, despite having recognised degree-awarding powers. A search for pre-compiled lists provided a lower count of individual institutions than our list: QS (https://www.topuniversities.com/university-rankings/world-university-rankings/2022): 90, UniRank (https://www.4icu.org/gb/public/): 146, The Uni Guide (https://www.theuniguide.co.uk/about/universities): 131, The Complete University Guide (https://www.thecompleteuniversityguide.co.uk/league-tables/rankings): 130, The Guardian Best UK Universities (https://www.theguardian.com/education/ng-interactive/2021/sep/11/the-best-uk-universities-2022-rankings): 121. This gave reasonable assurance that the list of URLs was comprehensive enough to provide an adequate overview of the current range of institutions.

With a list of target URLs created, the OAI-PMH protocol was employed to poll each URL to attempt to discover software records within the repository, as described in the following section.

### Querying IRs using OAI-PMH

Our initial approach focussed on automation using the OAI-PMH protocol to examine the presence of software records within the selected repositories using the URL list built in the

previous 'Data sources' subsection. To examine each repository for the presence of software as a defined set the OAI-PMH query `ListSets` was made against all URLs using Python 3.10 with the PyOAI library (*Infrae, 2022*). The response was tested for '`Type=Software`' or '`setSpec:74797065733D736F667477617265`', indicating that software had a predefined type within the repository. These set descriptions were discovered through manual analysis of those IRs containing software. To poll each repository for a list of records under these sets the OAI-PMH query `ListRecords` was used with the above `SetSpec` argument. However, around a quarter of servers responded with errors (*e.g.*, BadVerbError) to seemingly correct queries. Such URLs were listed separately and manually tested with WGET and a parameterised query string (*e.g.*, <URL>?verb=ListSets), with the responses parsed for the same information as the other URLs. Those URLs that still displayed errors using this secondary method were tokenised to get a base RIS URL (*e.g.*, pure.qub.ac.uk) and manually searched. These provided the fields 'contains_software_set', 'num_sw_records' and 'error' in the dataset.

This overall approach proved to be a poor indicator of software records when tested, with repositories known to contain software not having a software set in their list, thus preventing true records of software from outside that set being returned. Further, very high numbers of records were sometimes found in repositories that were known to not contain software. For example, one returned over 50,000, when a manual test showed it did not contain *any* records of type software. When the records were examined, it appeared that deleted records were also returned, which may include test data from the repository development. With a discrepancy between ground-truth data and the data yielded from OAI-PMH searches, we decided to perform a full manual test through the online search website of each repository to provide a ground-truth dataset. The full set of URLs with query strings has been made available (*Carlin, 2023*). While time consuming, this also helped to alleviate peculiarities. For example, one repository did not have a checkbox or selector for software, but inputting '`type = software`' to the search field revealed 23 records, which would have been missed otherwise. This provided the fields 'Manual_Num_sw_records' and the resultant 'Category', which categorised the IR into 'Contains software', 'Does not contain software' (*i.e.*, it has the functionality but no entry) and 'No direct software search capability' for IRs that do not categorise software as a distinct research output.

## Additional variables

Along with general descriptive metadata, the data retrieved using the CORE API also included details of the RIS platform software and the metadata format used, providing the fields 'ris_software' and 'metadataFormat'. A series of additional variables were added to the main dataset to investigate potential correlations between these and the presence of software. The 24 Russell Group universities were marked within the dataset as a dichotomous variable 'Russell_member' using the list available at *The Russell Group (2022)*. While the RSE movement has certainly grown rapidly within UK universities, it is not clear as to any effects this has had on the presence of software within UK academic repositories. To test this, the RSE groups mapped by the UK Society of Research Software

Engineering (*Society of Research Software Engineering, 2022*) were manually added to the dataset under a dichotomous 'RSE_group' variable.

## RESULTS

In this section, we present the results of the census into the prevalance of research software in UK IRs. We further test for relationships between software's presence in the IR and several descriptive variables of the IR, including its RIS framework, metadata format (OAI DC vs RIOXX), membership of the Russell Group and the presence of an RSE group in the institution.

### The extent of software in UK Academic IRs

To examine the overall prevalence of software in the repositories of UK academic institutions, the server responses were categorised into three subsets:

1. **Does not contain software:** those that contained software as an explicit type of entry in the repository, but did not have any records matching the type *i.e.*, capable of containing software as a specified category, but doesn't have any yet.
2. **Contains software:** those that contained software as an explicit type of entry in the repository and also have one or more records matching the type *i.e.*, can and does have defined software records.
3. **No direct software search capability:** does not have a specified entry type of software or similar *i.e.*, cannot contain an explicit entry of type software. Note, this does not mean there is no software in the repository, as each repository may have catch-all categories, such as 'other' or 'non-categorised'.

As depicted in Fig. 1, over 63% of the 182 polled IRs cannot contain software as a defined type. Less than 28% contain software and the remaining 8.8% have no records containing software, but are capable of categorising them as such. As each institution can have more than one repository, this data is re-presented in Fig. 2 to restrict one category per institution. For example, if an institution has two IRs, one containing software, then the institution is labelled as containing software. This analysis shows a similar order as per-repository, with an increased presence of software capability to an approximate 60-30-10 split.

Figure 3 shows the software records per institution, of those that do contain software. The percentage of software records is heavily weighted towards the top 10 institutions ranked by percentage contribution to the overall corpus of software records. For example, the top ranked institution contributes just under a fifth of all software records, with the top five responsible for just over half (50.93%). Among those repositories with software, the average number of software records was 30.24, though only 11 of 50 IRs were above that.

### Software records per research information system type

The brand or type of RIS was examined as a possible explanatory variable for the presence of software in the repository. The repositories were grouped by the overall framework, including all versions (*e.g.*, EPrints = EPrints3, EPrints3.1 etc.). The count of repositories using each type of framework is given in Table 1. The majority of repositories used the open-source EPrints software (57.14%), approximately 31% was made up of DSpace

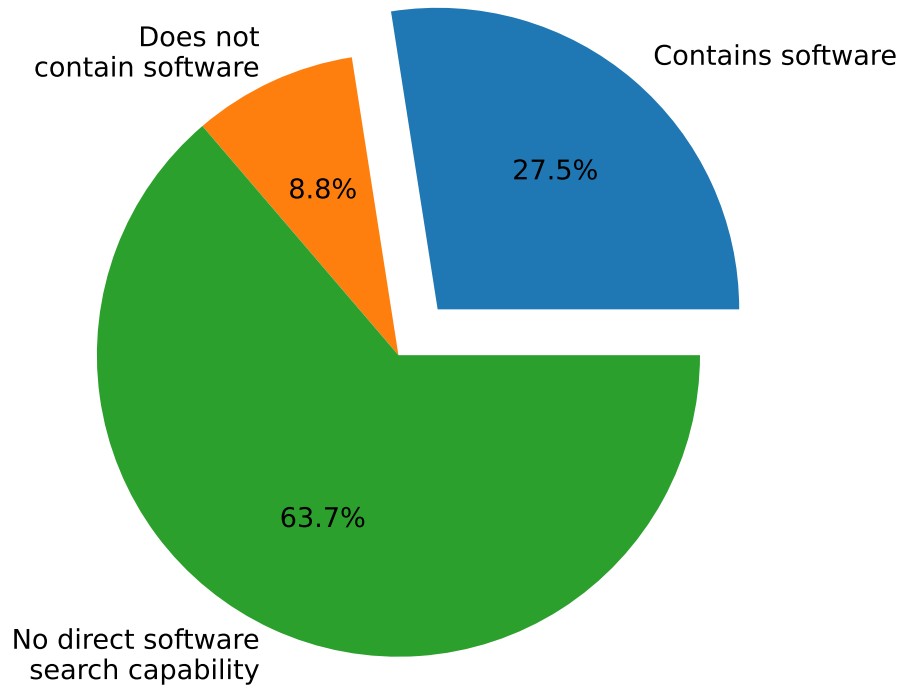

**Figure 1** **Software as a defined type in all repositories queried.**

and Pure, and the remaining seven platforms accounted for just under 11% of total repositories combined. As depicted in Fig. 4, of the ten types reported, half had no software record capability. To examine independence of the two variables, a Chi-square test of independence was performed using cross-tabulated categorical variables 'software records' and 'RIS framework'. This failed to reject the null hypothesis that both variables were independent of each other: $chisq = 9.048$, $p = 0.9586$, $dof = 18$.

## Metadata formats

When the dataset was examined for metadata formats, 106 entries used the OAI_DC format and 63 used RIOXX, with 13 missing values. To examine whether either format was correlated with software records being present, a cross-tabulation was conducted with these two variables (see Fig. 5). A Chi-square test of independence was performed using the cross-tabulated categorical variables, which failed to reject the null hypothesis that both variables were independent of each other: $\chi^2 = 0.0608$, $p = 0.9701$, $dof = 2$, indicating software records and metadata type are independent with $p > 97\%$.

## Russell Group membership

Membership of the Russell Group was cross-tabulated with the presence of software records in the dataset for all institutions, shown in Fig. 6. Russell Group universities were split

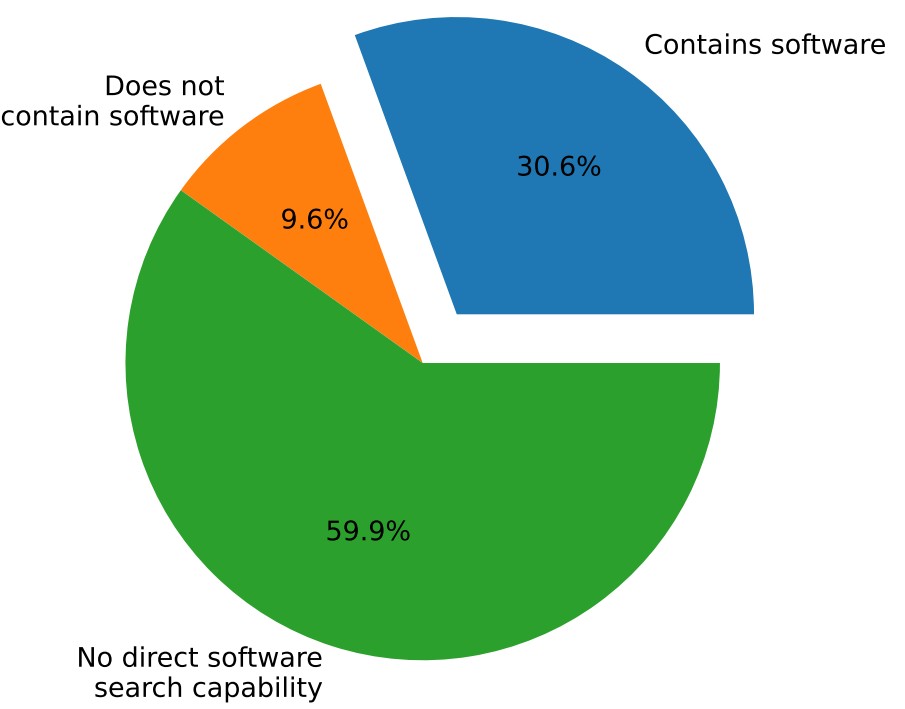

Software contained in
UK Academic Institutions

Does not
contain software

Contains software

9.6%

30.6%

59.9%

No direct software
search capability

**Figure 2** **Software as a defined type per institution.**

**Table 1** **Number of repositories per RIS type.**

| RIS type | Count | % |
|---|---|---|
| eprints | 104 | 57.14% |
| pure | 32 | 17.58% |
| dspace | 26 | 14.29% |
| worktribe | 6 | 3.30% |
| figshare | 5 | 2.75% |
| haplo | 4 | 2.20% |
| equella | 2 | 1.10% |
| QAICat | 1 | 0.55% |
| exlibris | 1 | 0.55% |
| fedora | 1 | 0.55% |

approximately 42% containing software, to 58% without and no instances of the type being included without records. Conversely, non-member universities had a 24.5% to 65% split, and over 10% with the ability to store a software record but none recorded. However, a Chi-square test of independence was performed using the cross-tabulated categorical variables, which failed to reject the null hypothesis that both variables were independent of

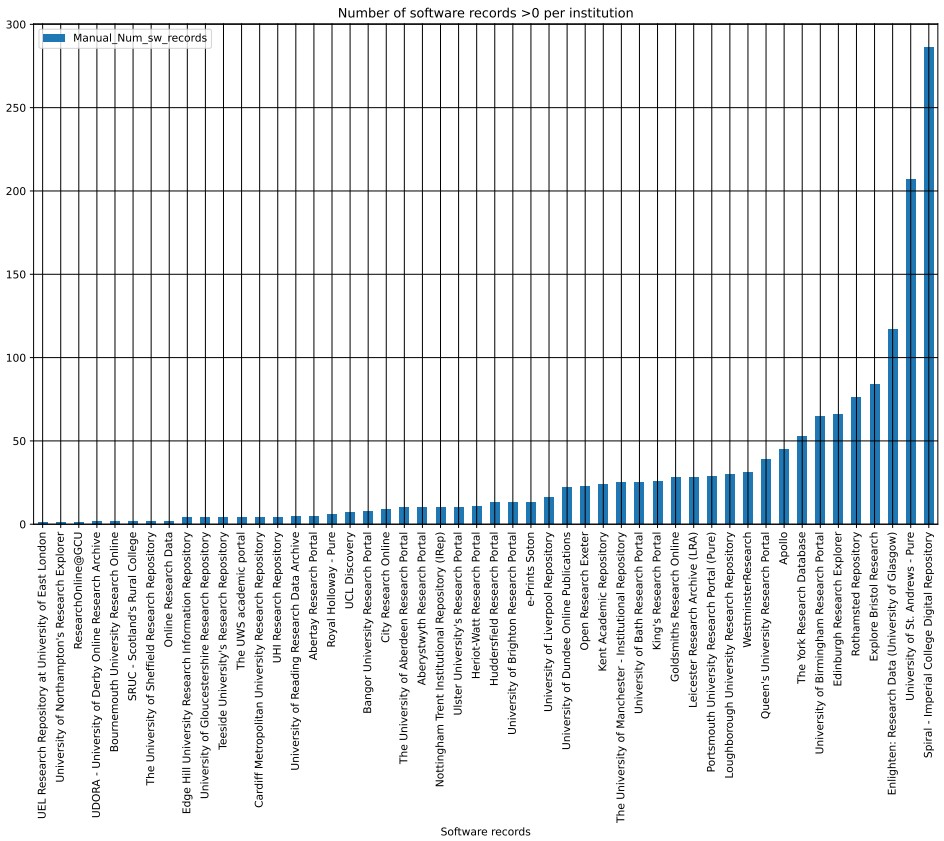

**Figure 3** Software records per institute containing software.

**Table 2** Proportional cross-tabulated RSE-group at institution with software records.

| RSE Group at institution | Contains software | Does not contain software | No direct software search capability |
|---|---|---|---|
| no | 0.225 | 0.106 | 0.669 |
| yes | 0.450 | 0.025 | 0.525 |

each other: $\chi^2 = 0.1555$, $p = 0.9252$, $dof = 2$, indicating software records and membership of the Russell Group are independent with $p > 92.52\%$.

## RSE Group at institution

The proportion of universities that employed RSE groups having software in their institutional repositories was twice as high than universities without RSE groups (22.5% vs 45%). However, more than half (52.5%) of repositories at a university containing an RSE group still had no software record capability. For a non-RSE institution, this was 66.9%. The cross-tabulated results can be seen in Table 2.

The independence of these two categorical variables was again tested using a chi-square test of independence, which failed to reject the null hypothesis that the two variables were
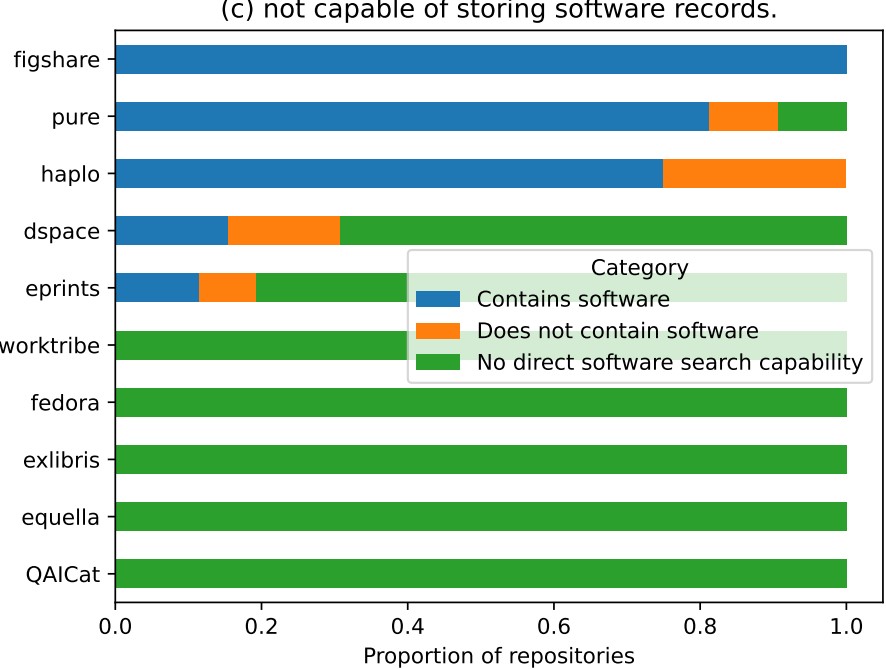

Proportion of repositories that (a) contain software records,
(b) contain no software records but are capable,
(c) not capable of storing software records.

**Figure 4** Proportion of repositories for each framework that contain software records, contain no software records but are capable, not capable of storing software records.

independent ($chi2 = 0.142$, $p = 0.9315$, $dof = 2$), indicating software records and RSE Group co-located at institution are independent with $p > 93.1\%$.

## DISCUSSION

### RQ1: to what extent is software published in academic repositories?

It is quite clear from the results presented in this research that there is a concerning lack of software stored in institutional repositories in the UK. More than 72% of the IRs polled had no software as a defined type within their repository and over 63% could not include it as specific output. When institutions, as opposed to repositories, were examined, nearly 70% had no software deposits. While it is certain that some research software will be held in other forms (personal GitHub/Lab, Innersource, private lab storage), this limits the reusability of the research code. Furthermore, a service like GitHub is a privately owned business, with no guarantee that it would not disappear tomorrow. While UKRI does not specify exact repositories in which software vital to the research should be deposited, it should fall under FAIR principles and essentially be governed by the same standards as a full journal article. It is more difficult to replicate and reproduce results (that are likely published in a more traditional format) if the code needed to produce and/or analyse the data is withheld, or obscured. Since most research is funded via taxpayer money, funders decree that everyone should have access to the data and resultant code. The enterprise and business development
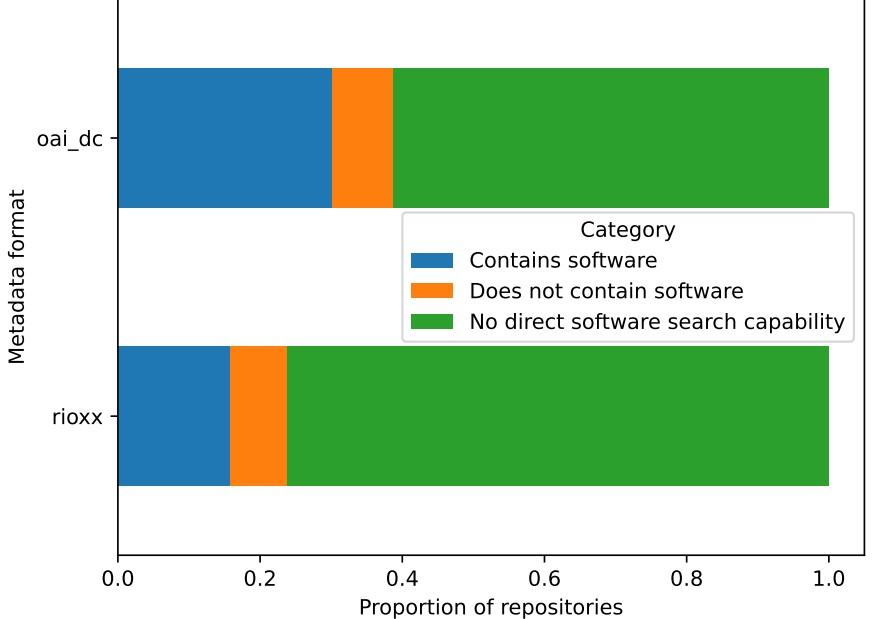

**Figure 5** Proportion of repositories for each metadata protocol that contain software records, contain no software records but are capable, not capable of storing software records.

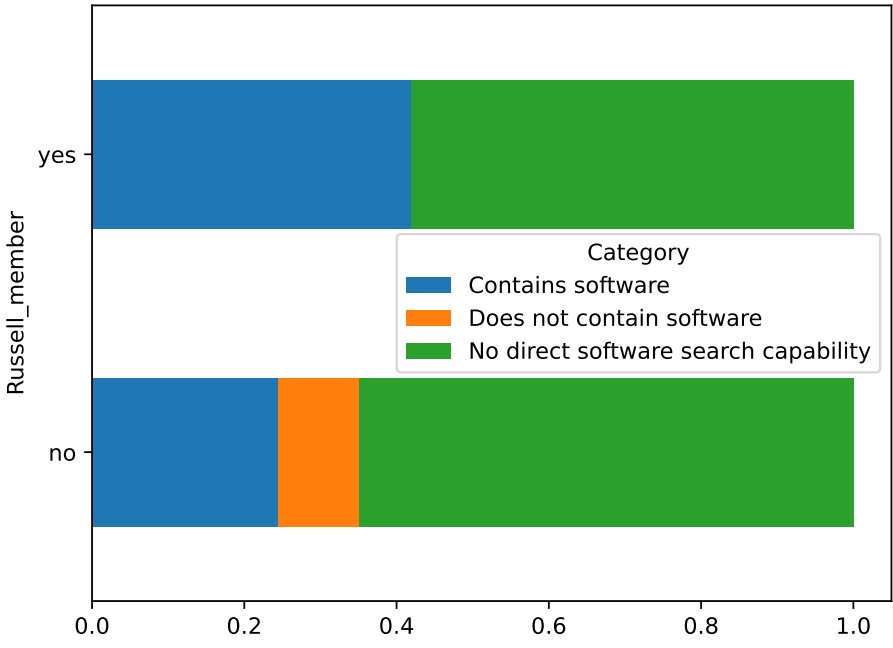

**Figure 6** Crosstabulation of membership of Russell group with software records in repository.

teams in universities may wish to exploit potential Intellectual Property (IP), which is well within the remit of universities. However, this can be easily managed with licensing and the fact that records in the institutional repositories do not need to contain the code itself, but even a pointer to the code location would be an improvement. This would operate as a code 'registry', but with the benefit of being maintained within the institution, non-volatile and respecting research software as legitimate research output, worthy of institutional archiving.

### RQ2: are there any explanatory variables associated with whether software is included in a repository?

None of the examined traits of the repository: RIS platform, metadata format used, RSE group in the same institution or Russell group membership showed correlational relationships with the presence of software in the repository. Certain RIS platforms do not contain software as a searchable item for any of the instances stored, though numerically these are not popular when compared to the three main RIS platforms, thus not impacting the result of this analysis. The three most popular open-source platforms (Pure, DSpace and EPrints) have instances in all three categories of our analysis, possibly due to the ease of changing configurations in the open source platforms.

In theory, metadata format could play a role in whether an IR contains software due the self-limiting nature of not having software as a type within the metadata vocabulary. While initially unexpected, our result indicating that metadata format does not correlate with the presence of software in an IR can be explained with some further analysis. Firstly, the source of the data indicating the metadata format was derived from the CORE dataset on data providers, not the individual institutions and this data is possibly incomplete. Many OAI-PMH servers offer multiple metadata formats in their responses, which can indeed affect the detail in the response. For example an OAI-PMH query (https://kar.kent.ac.uk/cgi/oai2?verb=GetRecord&identifier=oai:kar.kent.ac.uk:82465&metadataPrefix=rioxx) requested in RIOXX format to a known software record (*i.e.*, listed as such on the public-facing webpage (https://kar.kent.ac.uk/82465/) lists the type as: `<rioxxterms:type>other</rioxxterms:type>`, with no indication it is software. However, changing the metadata format to oai_dc in the OAI-PMH gives the detail sought: `<dc:type>Software</dc:type>`. Further, when the examining the source code of the item's webpage, it appears to present the information in DC format: `<meta name="DC.type" content="Software" />`. This example IR offers seven different options for metadata format, each with varying descriptors, though in the CORE dataset its format is listed as RIOXX. We consider this a limitation of the current study and future work could provide more granular data on the metadata types, particularly when public-facing webpages are used for the query rather than OAI-PMH.

The presence of an RSE group at the institution was investigated as a potential correlating variable as it may indicate the prevailing attitude to the importance of research software at the institution. This did not correlate with the presence of software. Large academic institutions are far from homogeneous in their views, processes and decision making. The decision to include software as a type of research output within a repository (or

wilfully omit it) could well be taken in complete isolation from other professional services functions within the institute, with no relation to the willingness to invest in professional research software engineers. Similarly, membership of the Russell Group of leading research universities did not correlate with software being present in their repositories. Again, this could well be symptomatic of the planning and management of the IR being entirely detached from other aspects of the university.

### RQ3: are there barriers preventing repositories from storing records of software as distinct research outputs?

There are likely many interacting contributing reasons for the lack of software in IRs, which we categorise as technical infrastructure, institutional and cultural in wider academia.

### Technical infrastructure

When the three most common repositories are examined (EPrints, DSpace and Pure), both Pure and DSpace contain software as an explicitly activated type by default. DSpace v7.4 has a default configuration that allows for software as a type (*Dspace, 2022*) and mappings to the DC terms for type. However, in our analysis, over 69% of the DSpace instances did not have software as a defined type in their search. Software as a type may have been purposely removed as a type of output, whether as an individual item, part of a larger reduction of items or a whole replacement of the default configuration. This is despite the output type's explicit definition within the DCMI (*DCMI, 2020*). Future work could investigate this or other potential reasons for the omission, as our data cannot provide definitive analysis. Pure has the default type 'software' under 'non-textual form' according to internal Elsevier support documents for Pure admins. The majority of Pure instances we polled have software as a searchable type, however we observed a case where it appeared to have been removed for that instance, suggesting again that it is an institutional and cultural issue, rather than technical. EPrints, which accounts for over 60% of repositories, does not have a default type for software. In communication with the development team, they elaborated that it is trivial to add new types of output, with a word in a configuration file (*EPrints, 2022*) enabling the type. However, even if institutional repositories include software as a type, CORE (the UK's repository aggregator), does not currently ingest this specific information and so it is lost in aggregation. This may change in the near future, with the updating of the RIOXX metadata protocol, upon which CORE relies, to include software. Given the default or trivial amendments needed to enable software as a type, we must consider other non-technical reasons for their omission or removal.

### Institutional

Academia can be slow to change long-held habits, with peer-reviewed journal articles still the prevailing currency for credit and career progression. For this reason, software can often be released along with the results in a traditional journal or conference article, sometimes with directions from the authors for users to cite their article, not the software. Thus the line between the article and the software becomes blurred. Specific journals have been established to make this process easier for those who write software, not articles (*e.g.*, Journal of Open Source Software). While benevolent, the focus again switches to an article

with a tacit acknowledgement of its status. The use of an article as the output to describe a secondary output (*i.e.*, the software) will likely explain some of the occurrences of software being omitted from the IR. Future work could examine this using text-mining approaches to determine the extent of this practise.

### Academic culture

It is entirely possible that a degree of intransigence exists around accepting software as valid scientific output in its own right, despite the firm mandate it receives from funders, policy-makers and researchers themselves. With the rate of software being submitted to REF dropping 90.2% to a mere 11 submissions since 2008, even if the infrastructure is in place to accept software, institutional norms may still hamstring the return rates regardless. There are many recommendations for hybrid pairs of repositories for different purposes, but if software and datasets are valid scientific output on a par with journal articles etc., why hold records of them in different, potentially unofficial, repositories? It would also appear that researchers who do deposit their research code in the likes of GitHub do not mirror this with entries in their institutional repositories, even when available. While the quantity of research software repositories on public hosts like GitHub may be hard to enumerate, several papers reviewed here have discussed growth in such hosting, with strong recommendations on the positive aspects. This trend may also explain, in part, declining institutional repository submissions of software.

## CONCLUSIONS

Major stakeholders in the research ecosystem underline the importance of research software to open science, but there is no current guidance, similar to that of a journal article, as to *how* this code is stored. While current RIS platforms do not have the flexibility to meet requirements for active development and maintenance, they can provide persistence to the records within the perimeters of the academic institution's own systems. Similarly, they are unlikely to provide neat cross-authorising integrations, like GitHub and Zenodo, to provide a seamless workflow. However, even with this workflow, over two thirds of the polled institutions could not accept software as a discrete unit of output. Institutions should be aware of the mandatory requirements of the major funders regarding software as an essential cog in the research process. It appears that some universities are intentionally removing software as a type of output, despite being in the default settings of the RIS. Only when these technical issues are resolved, can good software publication practices really flourish. Without this, the quantity of software as returns in the next REF might further decline, even to the point of extinction. While the submission requirements for REF exempt all non-textual outputs from the requirement to deposit the work in the IR within a given time-frame, they also mandate that REF-returned software is made publicly available. However, they do not define a scope for meeting this criterion satisfactorily.

## RECOMMENDATIONS AND FUTURE WORK

Having examined the current state of software within academic institutional repositories, we offer some recommendations for solving the prevailing issues preventing software being

accepted as a type of research output. Firstly, clear standards should be set for expectations around the use of institutional repositories as permanent records of scholarly output. Secondly, it is trivial to enable software as a type of output, with a single word entry in the config file of the most popular RIS platform being enough. We also suggest that software should be enabled by default on the EPrints system through this method. Finally, to 'square the circle' between development needs, persistence requirements and institutional records, the institutional repository should have an easy workflow to mirror records from development repositories (*e.g.*, GitHub) and code-friendly archival repositories (*e.g.*, Zenodo) within the institutional realm. We propose further development of potential lightweight integrations to make this workflow seamless across major RIS systems. An ideal scenario would be an institutionally hosted development-first repository, with persistence attributes through DOI minting and direct mirroring of the record in the institutional repository.

Future work could examine the links between 'shadow', semi-official and official off-site repositories, with institutional versions. Future policy initiatives can lobby for institutional-level change, making requirements for the archiving of research software. Lastly, this work focussed solely on UK academic institutions. With the rapid growth of the RSE movement worldwide, it would be beneficial to compare this work to results from around the world, particularly notable RSE communities in the USA and Europe.

## ACKNOWLEDGEMENTS

We would like to extend our gratitude to the editorial team at PeerJ for helping to shape our manuscript into the published article.

### Funding

This work was supported as part of Dr Carlin's UKRI EPSRC Research Software Engineering Fellowship (EP/V052284/1). The funders had no role in study design, data collection and analysis, decision to publish, or preparation of the manuscript.

### Grant Disclosures

The following grant information was disclosed by the authors:
UKRI EPSRC Research Software Engineering Fellowship: EP/V052284/1.

### Competing Interests

The authors declare that there are no competing interests.

### Author Contributions

- Domhnall Carlin conceived and designed the experiments, performed the experiments, analyzed the data, performed the computation work, prepared figures and/or tables, authored or reviewed drafts of the article, and approved the final draft.
- Austen Rainer conceived and designed the experiments, authored or reviewed drafts of the article, and approved the final draft.

- David Wilson conceived and designed the experiments, authored or reviewed drafts of the article, and approved the final draft.

## Data Deposition

The dataset is available at Zenodo and at the QUB Institutional Repository:

- Domhnall Carlin. (2023). A census of research software in 171 academic institutional repositories. (Version 1) [Data set]. Zenodo. Available at https://doi.org/10.5281/zenodo.7603444

- Carlin, D. (Creator) (03 Feb 2023). A census of research software in 171 academic institutional repositories. Queen's University Belfast. submission_dataset_release_v1(.csv). 10.5281/zenodo.7603444

The code is available at Zenodo: Available at https://doi.org/10.5281/zenodo.7974836

The dataset of manual queries is available at Zenodo: Carlin, Domhnall. (2023). institutional_repo_census (v1.0.0). Zenodo. Available at https://doi.org/10.5281/zenodo.7974836

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
