# Peer review of "Where is all the research software? An analysis of software in UK academic repositories"

_PeerJ Computer Science, doi:10.7717/peerj-cs.1546_

## Round 0.1 · original submission · Major Revisions

Both reviewers agree that there are a lot of useful results in this paper, and encourage further development.

However, they also highlight difficulties in understanding the narrative and context of the paper, and would like to see a more thorough analysis of the results. It would also be useful to ensure that the paper is understandable to people from a non-UK background - many countries do not have exercises like REF, but do have an interest in understanding how best to support policies around software in institutional repositories.

Both reviewers also agree that your methodology, while appropriate, could be better documented.

Please could you revise your paper to address all the reviewer comments.

Overall, although this is marked as major revisions, I do not think that there is a requirement to change / add to the experiments that you have done. This decision is marked as major revisions as rewriting will be required for most sections in the paper, in particular ensuring that the experimental design is documented in a way that ensures readers can trust the results.

Reviewer 1 ·

Basic reporting

The authors use clear language throughout the article. I found the introduction especially compelling and thought that the authors did a good job positioning research software as an artifact in need of professional recognition and as important for reproducibility. However, the article would benefit from the addition of more signposting and meta-narration to help the reader as well as some restructuring.

I suggest this in part because I found myself wondering multiple times why something was relevant (only to find out later) or why details were provided in one section but not another. For example, it is unclear why RIOXX is important until the results section. And, while some variables were introduced in the first half of the paper, Russell Group membership is not introduced until the results section. Additionally, I recommend these changes because the methods used were somewhat confusingly presented; substantial attention was given to an approach that seems never to have been used and some details feel missing. These confusing aspects contrast significantly with the very approachable introduction and the inclusion of fundamental information like what software is.

I recommend making some content cuts, reorganizing some earlier sections, and adding to the methods section. Cut superfluous content like the definition of software (your readers know this) and functions of scientific communication (this is too perfunctory a discussion to be meaningful). Instead, begin the background section with a summary of why the details provided there are important. Why does the reader need to know about the Dublin Core? What does this have to do with the design of the study at hand? Rather than make them guess, briefly tell them at the start of that subsection why the topics summarized are important. I imagine this would involve something to the effect of, "Here we present background information on a set of variables we anticipated might be related to the amount of software deposited in a given IR." Including an explanation like that would clarify the study design, help the reader understand the purpose of the content, and give more structure to the paper. I further recommend combining the background and related works sections as the distinction between them is unclear. For example, the distinction made in the background section between registries and repositories recalled the distinction made in the related works section between a code and an archival repository. I would also add background on OAI-PMH here rather than in the methods section; this will be more consistent with the way other metadata tools like RIOXX were treated.

With a shorter background/related works section, there will be some useful room to expand the methods section. I recommend adding some more structure to the methods section. Again, a summary of the section's contents up front would be useful. Then present the methods in chronological order with the relevant details. I recommend reorganizing the methods section so that you clearly present all variables collected and describe how this was done (the reasoning for why will have already been established in the background section). You seemed to 1) identify a list of UK IRs; 2) use metadata accessible by OAI-PMH to check that list for IRs containing software; 3) validate results manually, finding major discrepancies; 4) gather data on other variables like Russell group membership, RSE group presence, etc. After presenting this/a similar summary, go into detail on each step.

As is, the methods section provides technical detail but seems to assume important details. You created a list of IRs and then how did you use OAI-PMH to harvest metadata? How many items were in the list? Why would you test for setSpec: 74797065733D736F667477617265? What does it mean that a large amount of errors were returned? I was also confused by OAI-PMH’s sets—what does it mean for a repository to have “a defined set”? What is the significance of the sets column in Table 2? What is a base RIS URL? Why were your variables chosen? Furthermore, the methods section dwells on the use of OAI-PMH but its ultimate role in the research is unclear to me—the multiple figures dedicated to this tool further muddy my understanding. I think that moving background information out of the methods section and more clearly presenting the methods in a step-by-step way will help ensure that (only) necessary information is shared here.

Signposting at the beginning of the results section would also be useful, e.g. "We test for the relationships between software’s presence in the IR and several characteristics of the IR including its supporting software framework, metadata format (OAI DC vs RIOXX), membership in an elite research group...." Figure 4 could be cut; it feels unnecessary after Figure 3.

Regarding the discussion: as it is, it feels under developed. Rather than postulate that technical infrastructure, academic culture, and institutional factors affect IR use for software, why not answer your research questions directly? The discussion section could have subsections to address each of the RQs. I don't think this is the only possible approach to improvement, but it would mean that the RQs don't get abandoned. At the very least, a stronger tie to the literature and the study's results is necessary for the discussion. The content now seems like tenuously related speculation, especially because there are few clear connections to the results (does the Institutional section have any direct relation to the results?).


A few smaller things:
The acronym RIS seems to get used without definition.
Line 445 has an incomplete sentence: “The proportion of each”
Errant comma on line 468
“The likes of GitHub” is odd phrasing for a publication
Sentence starting on 488 is maybe missing a verb? Something is weird there.
491—what is “this”?
507 - should have comma not dash after CORE
Missing period line 511
Inconsistent use of comma after i.e.
Are the pie chart colors distinguishable for color blind people?
Typos in abstract. Should read “but there has been no” not “but there as been no.” Should read “changes in policies and operating procedures” not “changes in policies an operating procedures.”


The data and code for the study were shared.

Experimental design

A major question I think needs answering is how the list of UK IRs was developed. Why should I believe this is a complete list with no redundancies? Much like establishing the search parameters and database choices in a systematic literature review, this is fundamental to interpreting the study's findings.

I recommend making the variables studied and the reasoning for their choice clear in the background section.

Otherwise, the design and of the study is appropriate, though not completely clearly communicated. As detailed above, the steps taken to collect and analyze data are slightly unclear and the reasoning behind variable choice is not specified. Research questions were put forth and not explicitly returned to. RQ4 is not effectively answered through a study of this kind and should probably be removed and only addressed as a discussion point.

Validity of the findings

Data are provided and appear to produce the same results presented in the paper. A codebook for the data would be a welcome addition as would a step-by-step of how to run the python scripts to produce the dataset. The Github README references create_base_data_set.py but it doesn’t exist in the files. Perhaps you mean build_main_dataset.py?

The discussion could better address the relationship between the study's findings and those in the literature.

Additional comments

The contributions of the article are great! I'm fascinated by IR's removal of software as a type and the low use of IRs to store software. However, I do think the telling of background, methods, and discussion could be improved to improve the flow and more strongly relate findings to discussion. There needs to be more justification for the variables chosen in the study and the approach to establishing a list of IRs. Because some of these requested changes relate to establishing the study design as appropriate, I think this warrants a major revision.

·

Basic reporting

The basic reporting is OK,
except that the raw data is not shared.

You did a manual search (without using OAI-PMH). Even for this, I suggest to make the search results available as supplement.

Experimental design

no comment

Validity of the findings

Thanks for your study, it is interesting to see such a steep decline in numbers of research software submissions to the REF.

I think that your concrete recommendations at the end of the paper are valuable.

However, your observations have its value, but I would like to see some more attempts to interpret the results.

First, please note that I’m not affiliated in the UK, thus I’ve no detailed knowledge of the REF evaluation criteria.

In your paper, you should explain how much the REF ranking is influenced by the number of research software publications. This my help to interpret the results.

I assume that a lot research software is published at GitHub and GitLab. With snapshots at Zenodo, as you also mention in your paper. Do such publications count for REF? If yes, UK universities have no incentives to (also) publish in institutional repositories. May be, this is an explanation of the decline?
You also discuss this on Lines 529-532 (I consent with this discussion).

You mention that “Use infrastructure that minimizes the risk of future access restrictions or control by commercial organizations. (BOAI, 2022)”.
This applies to GitHub, but not to Zenodo, which is operated by CERN.

You write that “Archival repositories provide accurate long term immutable references to any form of digital object, tracked by a permanent DOI.” Zenodo provides this.

You write that metadata is crucial in making software FAIR (Line 208). That’s true, and Zenodo does not provide specific support for research software. Do institutional repositories provide specific support for research software (not just the type of entry as you discuss for eprints)? This would be interesting for this paper. You mention the Dublin Core (Line 443 ff), but this is not specific for software. Which metadata would be appropriate? Do standards such as Codemeta or the Citation File Format (CFF) help?

Additional comments

In Section “Software: a definition”, I suggest to discuss the distinction between “research software” and “software in research”, see (Chue Hong et al. 2021).

Line 126: The Journal of “Software: Practice & Experience” does *not* peer-review research software and publish it. It should not be listed here.
Instead, you may list Elsevier’s “Software Impacts” journal.

Typos:

Abstract: … there Has been …

Line 111: of of

Line 133: (DORA) (DORA)

Fig 3 and 4 are too small to be readable.

---

## Round 0.2 · accepted · Accept

Please note that there are two typos to correct before publication:

P9, Line 446. Missing space after period on "157 institutions.However"
P15, Line 586-587 use a parenthetical to list the traits—the colon doesn’t work "traits of the repository (RIS platform, [...] or Russell Group membership) showed"

Please also check for any additional spaces inserted before a footnote reference.

Reviewer 1 ·

Basic reporting

In its first draft, the article was clearly written, cited appropriate references, described relevant background, and reported relevant results. However, as detailed in my initial review, the draft could be improved with more meta-narration, rearrangement of some content, and explicit responses to the RQs. The authors carried out these suggestions well and I think their readers will appreciate it. This revision is a good addition to the literature (and interesting to me personally!).

Experimental design

The experimental design and research questions were appropriate. The revision has made the data collection process more clear and introduced variables earlier in the text—an asset for comprehension.

Validity of the findings

Data and code are provided. The conclusions are more directly linked to the RQs in this revision, thank you!

Additional comments

I put minor revisions only because I noted two typos/grammatical errors. I happily accept this article otherwise and would not need to re-review. The errors I noticed are:
Line 461, p9, Missing space after period on
Line 628, p15, use a parenthetical to list the traits—the colon doesn’t work

·

Basic reporting

The basic reporting is OK now, thanks!

Experimental design

Properly designed and presented.

The review comments are addressed, or reasons for not addressing them given.

Validity of the findings

Thanks for publishing the software (notebook) and the data sets!

Conclusions are appropriate.

Additional comments

none